# VARYING MEANING COMPLEXITY TO EXPLAIN AND MEASURE COMPOSITIONALITY

**Tom Bosc**
Mila, Université de Montréal
`bosct@mila.quebec`

## ABSTRACT

Compositionality is assumed to be a key property of language, but it is hard to observe in language emergence simulations. Following De Beule & Bergen (2006), we posit that the meaning of the datapoints that agents discuss must vary in complexity. We extend their work in different directions. First, we argue that this variation in the task is realistic and underlies the emergence of intersective adjectives and argument structure. Secondly, we show promising results for this hypothesis with attention-based neural networks. Thirdly, we argue that languages learned on tasks where meaning complexity varies are easier to analyse, and propose an intuitive metric called concatenability to illustrate this claim.

Natural language is often thought to be *compositional*, so that "the meaning of a complex expression is determined by its structure and the meaning of its constituents" (Szabó, 2020). However, we do not fully understand the nature of compositionality, how it emerges and how it can be integrated as inductive biases in artificial agents. For Zipf (1949), language maximizes information transmission while minimizing the effort of the speakers and listeners. This optimisation process is constrained by factors internal to the agents (memory constraints, preferences for iconicity, etc.) as well as by external factors such as the distribution of the meanings, pragmatics or discourse factors (Kirby, 1999). We can tease apart these factors using computer simulations of language evolution and acquisition (Steels, 1997; Lazaridou & Baroni, 2020).

However, such simulations are often not very successful, unless they use unrealistic assumptions, in which case they also lose part of their explanatory value. For example, Kottur et al. (2017) restrict the vocabulary size of the speaker to be exactly the number of different values to denote, and also erases its memory after each utterance. Korbak et al. (2019) proposes that compositionality evolves from simpler communication protocols where the primitives ("blue", "circle") are learned in a separate training process, but the syntax of the agents is hardcoded through very specific training procedure involving several agents.

The difficulty of quantifying compositionality is yet another hurdle to progress. When agents exchange variable-length utterances without an explicit model of grammar, it is not clear whether there are any meaningful subconstituents or not, and if there are any, how to segment them in order to measure the quantities of interest. Thus it is often implicitly assumed that each symbol bears meaning. However, this can lead to strange conclusions: if we assume that each letter in "cat" is meaningful, then English is not compositional.

To tackle these problems, we propose to study language emergence simulations where the complexity of the meanings to convey vary. Combined with a loss function encoding Zipf (1949)'s least-effort principle, we posit that grammatical structures will grow in complexity to accommodate for the variations in complexity.

A similar hypothesis was proposed by De Beule & Bergen (2006) and we build on their work in the following ways. Firstly, we argue that such variations are present in everyday communicative situations. We propose two very different setups to explain the emergence of intersective adjectives and argument structure. In this view, it is not a drawback of the method that task complexity must be low for compositionality to emerge, since it is realistic. Secondly, we show how to reproduce their conclusions using neural architectures. In their work, agents use explicit grammars, whereas ours use neural models based on attention mechanisms. Our work adds weak evidence for the learnability (and not only the existence) of systematic connectionist architectures, contra Fodor et al. (1988).

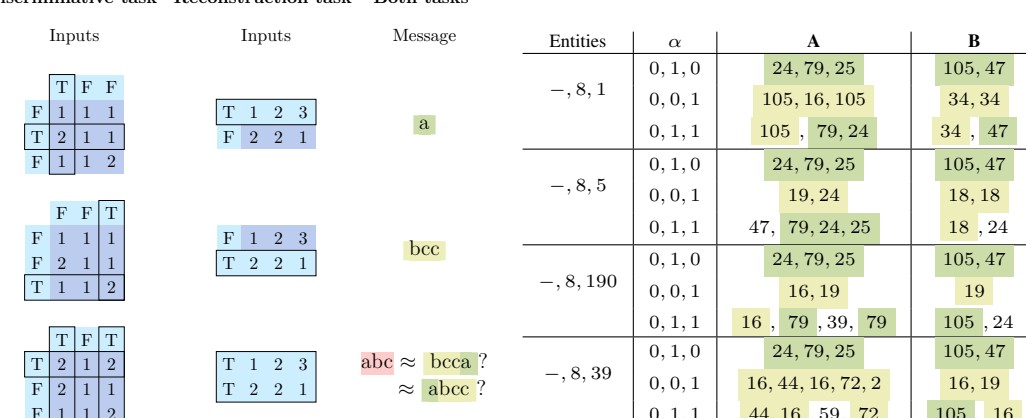

Figure 1: Simplified illustration of the two tasks. *Feature matrices*, where rows are objects and columns their properties. *Masks* contain booleans {T, F} indicating to the speaker what to communicate about (pragmatically relevant information). Both speaker and listener observe information colored in dark blue, but only speaker observe cyan. Our metrics (cf. Section 2) compare utterances (last column) about simple meanings (two first rows) to utterance about complex objects (last rows).

Table 1: Example messages from two models trained on the reconstruction task. Objects: list of objects ("-": no object; number indicate filler; position in the list indicate role: AGENT, PATIENT, MISC). $\alpha$: mask. **A**, **B**: messages produced by speakers from two different runs. Symbols are manually colored to identify phrases (first 2 rows in every block of 3 rows) within artificial sentences (third row in every block). We see that some systematicity has emerged.

Finally, we show how meaning complexity variation helps to study artificial languages. Having access to messages denoting objects or properties in isolation, we can sidestep the need to complex artificial messages into smaller constituents. This framework opens the door to a variety of novel metrics. We conclude with an overview of the opportunities afforded, outlining some concrete next steps.

# 1 TWO REALISTIC SETUPS WITH MEANING COMPLEXITY VARIATION

Compositionality is typically studied mathematically as follows (Montague, 1970; Szabó, 2020). Consider a meaning function $m$ that maps expressions taken from a language $\mathcal{L}$ (the space of grammatically correct expressions) to a space of meanings $\mathcal{X}$ (all the things that the language can denote). A language is compositional iff $m$ is an homomorphism, that is, if there are two binary operators, $\cdot$ on $\mathcal{L}$, and $\times$ on $\mathcal{X}$, such that for any expression made of two constituents $e_1$ and $e_2$ in $\mathcal{L}$, we have

$$m(e_1 \cdot e_2) = m(e_1) \times m(e_2). \tag{1}$$

In language emergence simulations, $\mathcal{L}$ is the set of variable-length sequences of symbols produced by an agent (the speaker). In our case, symbols are sampled from a vocabulary $\{1, \ldots, n_V\}$ and the length of the messages are bounded. The operator $\cdot$ is the concatenation operator, so we write $e_1 \cdot e_2 = e_1 e_2$.

The meaning space $\mathcal{X}$ contains sets of filler-role pairs, with the constraint that each role appears at most in one pair. As we will see, the fillers and roles differ slightly the two tasks. Since we use sets, we can take $\times$ to be the union on sets.[1]

We define *simple meanings* as sets from $\mathcal{X}$ containing a single pair, while *complex meanings* contain at least two pairs. A task with **meaning complexity variation** contains both simple and complex meanings to convey.

We assume a functional view of language seen primarily as a communication device, shaped by its use. The agents jointly optimize a loss function which has two terms, balancing the maximization

---

[1]This is an oversimplification: we need to ensure that each role appears at most in one pair.

of information transmission while minimizing the efforts of the speaker. This system is a partial, particular model of the functional view put forward by Zipf (1949).

We further assume that this objective is the cause of various pragmatic effects such as the maxim of quantity of Grice (1975), according to which the speaker convey information that is necessary to solve a given task, and nothing more. We found that inferring what part of the input should be conveyed is hard for current architectures and optimisation techniques. Therefore, we help the agents by providing pragmatic annotations as inputs. Agents can use these annotations to respect the Gricean maxim and communicate optimally, and they are encouraged to do so by a least-effort penalty, but they do not *have* to do so.

## 1.1 THE EMERGENCE OF COMPOSITIONALITY IN INTERSECTIVE ADJECTIVES

The first task is a discrimination task (Lazaridou et al., 2017; Havrylov & Titov, 2017). The speaker observes a set of objects from $\mathcal{X}$ and one of these is marked as a target, while the others are distractors. The speaker produces a message that characterizes the target. The listener observes the same set of objects and must pick out the target using the message. Each object $x \in \mathcal{X}$ is a set of attribute-value pairs. For example, a blue circle can be described as $x_1 = \{(\text{COLOR}, \text{BLUE}), (\text{SHAPE}, \text{CIRCULAR})\}$.

Crucially, the agents are exposed to simple meanings, for which a single attribute-value pair is necessary to pick out the target, as well as to complex meanings, for which two pairs are necessary (cf. first column in Figure 1). In other words, the meaning complexity varies.

This task simulates hypothetical conditions of emergence of intersective adjectives. In natural language, these adjectives are concatenated sequentially, as in "the *blue circular* thing".[2] There are several important parallel between frequent, real-life communicative situations and this task, but we need to focus on the realism of our innovation: the variable meaning complexity. The primary function of adjectives is probably to distinguish between *similar enough* objects. One does not use adjectives to discriminate between different kind of concrete things such as a car and a tree, as these things are more naturally associated to basic level terms, which probably have a lesser cognitive cost. Thus, in our task, some of the meanings are such that a single attribute-value pair is necessary to solve the task. By contrast, works from Lazaridou et al. (2017); Havrylov & Titov (2017) seem to sample objects independently.

To help the speaker to be concise and thus obtain more natural languages, we provide it with the minimal set of attributes that distinguish the target from the distractors. This set of attributes is given to the speaker via a mask over features, as illustrated in Figure 1 (column 1).

## 1.2 THE EMERGENCE OF COMPOSITIONAL ARGUMENT STRUCTURE

In the second task, meanings in $\mathcal{X}$ are *events* which contain entity-role pairs. We can think of these as representations of clauses. For instance, one such event would be $e = \{(\text{ABOVE}, x_1), (\text{BELOW}, x_2), (\text{REL}, \text{ABOVENESS})\}$, where $x_1$ and $x_2$ can be the geometric shapes of the previous task. To simplify the experiments and the analysis of the results, the pair with the REL attribute is observed by both agents and needs not be conveyed. The complete description of this setup will be published soon in a separate publication.

Crucially, each relation takes a variable number of roles. *Simple* meanings require a single entity-role pair to be conveyed while *complex* meanings require several. Furthermore, the listener observes a partial view of the input of the speaker. This is illustrated in the second column of Figure 1: a meaning made up of two entities can be partially observed (first 2 rows) or completely unobserved by the listener (last row). When all but one entity is observed by the listener, the meaning is simple, since only the entity hidden to the listener must be communicated. Otherwise, the meaning is complex. Thus meaning complexity varies across inputs in this task as well.

This task simulates the emergence of argument structure. Indeed, in all natural languages, the way arguments are composed seems to be highly systematic. In simple sentences, arguments are simply concatenated to the verb. This seems to be true, regardless if roles are encoded via word order (analytic languages), case marking (synthetic languages) or verb affixes (polysynthetic languages).

---

[2]Our setup does not explain if and how they are ordered.

The most important characteristic of this task is probably partial observability, and it corresponds to a fundamental aspect of the human experience. Most events that have happened in the past or will take place in the future are not directly witnessed and are only partially represented in our minds. Yet we often want to represent them more completely, which prompts one to ask questions, or spontaneously share information that is ignored by the interlocutor. For instance, upon seeing a broken window, one can ask "who broke the window?". A knowledgeable interlocutor could answer "John" or "John did". In our experiments, the speaker is this knowledgeable person, answering such questions about unobserved entities. Even if the agents do not engage in dialogues but in one-time interactions, the mask models an inference made by the speaker about the listener's knowledge, and abstracts away over all the possible reasons to share this information. Going back to our example, "John did" is an acceptable answer which does not contain the phrase "the window". This parsimony is encouraged by the least-effort penalty.

In Mordatch & Abbeel (2018)'s and Bogin et al. (2019)'s works, agents also need to express such events. However, the complexity of the meanings do not vary.

A formal and more detailed comparison between the two tasks is available in Section A.

## 2  SIDESTEPPING THE NEED FOR SEGMENTATION

To verify if compositionality holds as formalized in equation 1, we need to access the constituents $e_1$ and $e_2$ that make up the complex expressions of the form $e_1 e_2$. In some works, this structure is known (Andreas, 2018), but it is more typically assumed that every symbol is meaningful (for example, in Bogin et al. (2019) and Chaabouni et al. (2020)'s works). However, neither of these options seems appropriate to analyze the messages exchanged by neural agents. This problem is also identified by Baroni (2020), but to our knowledge, we are the first to address it, or rather to sidestep it.

Using tasks with meaning complexity variation, we can encode simple meanings $\{x_1\}$ and $\{x_2\}$ in isolation to obtain meaningful constituents $e_1$ and $e_2$. This is possible because some of the simple meanings are part of the training data. Let $e_{1,2}$ be the message produced by the speaker to discuss $\{x_1, x_2\}$. To quantify compositionality, and in particular, systematicity, we would like to measure the similarity between $m(e_1 e_2)$ or $m(e_2 e_1)$ and $m(e_{1,2})$ (cf. third column in Figure 1).

One way to do so is to compare the loss incurred by the listener when it interprets $e_1 e_2$ versus $e_{1,2}$. Formally, we define the information transmission loss (reconstruction or discrimination) incurred when the complex meaning $\{x_1, x_2\}$ is conveyed via $e$ as $l(e) = -\log p(\{x_1, x_2\}|e)$, where $p$ is the conditional distribution of the listener. Then, the *listener's concatenability* is defined as

$$C^L = l(e_{1,2}) - \min(l(e_1 e_2), l(e_2 e_1)). \tag{2}$$

Following (Lowe et al., 2019), we propose two metrics that capture the understanding of the two agents separately. We define $u(e) = \log q(e|\{x_1, x_2\})$, the log probability (according to the speaker's conditional distribution $q$) of an expression $e$ when the complex meaning is observed. The *speaker's concatenability* is defined similarly as

$$C^S = \max(u(e_1 e_2), u(e_2 e_1)) - u(e_{1,2}). \tag{3}$$

These metrics are positive iff the concatenation $e_1 e_2$ (or $e_2 e_1$) is preferred to the actually sent message $e_{1,2}$. For the speaker, this means that the concatenation $e_1 e_2$ (or $e_2 e_1$) is more likely; for the listener, that $e_1 e_2$ (or $e_2 e_1$) allows it to recover the information better.

The $\min$ operator picks the lowest reconstruction loss among the two possible concatenation orders. Thus, the method is agnostic to the order of the subexpressions, similarly to bosdis in Chaabouni et al. (2020)'s work. However, we emphasize that these metrics are only examples of what one can do with our framework. For instance, to study the importance of the order of the subexpressions, one could study the gap $|l(e_1 e_2) - l(e_2 e_1)|$. We believe many more insightful and intuitive metrics can be imagined when one has access to subconstituents $e_1$ and $e_2$.

## 3  EXPERIMENTS

The commonalities between the two tasks (inputs, targets and loss functions) are given in Section A. In all our architectures, attention mechanisms (Bahdanau et al., 2014) plays a crucial role of

selecting what information to send next. In particular, the models used are based on Transformers (Vaswani et al., 2017). The main difference between the architectures in the two tasks is that the sender attends over objects in the argument structure task, while it attends over properties of the target object in the intersective adjective setup. Since our experiments on the second task will be published soon as a separate publication, we focus on the first task, the discriminative task.

We use synthetic data in order to carefully control the complexity of the meanings, as described in Section B.1. The sender represents the target object as a set of vectors (one for each property), and produces a message by attending over these vectors. The properties that are pragmatically relevant to distinguish the target from the distractors are marked with a specific embedding. The listener embeds all the objects independently and represent the message in a single vector using a Transformer encoder block. Then, a probability over objects is given simply by computing the dot product between the message vector and all the objects representations. This process is detailed in Section B.2.

Once we have trained our runs and filtered out the runs that failed (cf. Section C), we perform a qualitative and quantitative analysis. To give an intuition about the concatenatibility metrics, we show messages produced by a sender-listener pair with a low $C^L = -1.19$ (model **A**) versus one that yields a high $C^L = -0.05$ (model **B**) in the discriminative task. We encode all utterances from the training set, compute the average $C^L$ and compute the most frequent message for each pragmatically relevant features (simple and complex). We report (non-cherrypicked) examples.

In Table 3, we see that **B**'s sender produces messages that are often the concatenation of messages for simple meanings in isolation. By contrast, **A**'s sender produces messages that are related to either or both messages, but the messages seem to be less structured and predictable. Thus concatenability seems to capture an intuitive definition of systematicity that 1) is correct regardless of the lengths of the expressions to compose and 2) is invariant with regards to the order of the sub-expressions. More details are provided in Section D.1.

# 4   DISCUSSION AND CONCLUSION

*Meaning complexity variation* is a simple and promising functional explanation for the emergence of compositionality. We have argued that in conjunction with the least-effort principle, it could be a common cause behind the emergence of intersective adjectives and simple argument structure. This explanation is orthogonal to other characteristics of the tasks (reconstruction vs discriminative tasks) and could operate at different linguistic levels (phrase-level, sentence-level).

In addition to the functional explanation, we obtain a whole class of methods to analyze compositionality (and in particular, systematicity), as well as the ability to apply existing metrics such as those proposed by Chaabouni et al. (2020). We think that it is not a coincidence if meaning complexity variation both explains compositionality and helps us to study it. The easier languages are to analyze for scientists and machines, the easier they should be acquired by children as well, a reasoning that resonates with Gopnik (1996)'s theory.

However, this double role makes it harder to verify that our hypothesis is different from two alternative explanations. The first alternative explanation is that what matters is that inputs have the "right" meaning complexity. For instance, does compositionality emerge when all examples are complex, or when all examples are simple? The second alternative explanation is that what matters is the variation in the *quantity* of information (i.e. the non-uniform character of the distribution of the inputs), not the structure or complexity of meanings. In our setup, the two go hand in hand. We need to verify such claims as rigorously as possible in the future.

We also need to study whether there is a relationship between generalisation and compositionality. For instance, Chaabouni et al. (2020)'s metrics are not necessary for generalisation. Is it also the case for concatenability?

Finally, the two tasks could be merged within a single simulation. The meanings of the argument structure task are made of several entity-role pairs, but the entities themselves are meanings of the adjective task. By training agents to solve these two tasks at once, one expects adjectives to start appearing within arguments. More generally, this opens the door to simulations where various parts of speech emerge.

ACKNOWLEDGEMENTS

We thank Pascal Vincent for his work on the preceding paper, NSERC for financial support, the Mila IDT team for the computational infrastructure, and the reviewers for their useful and thorough feedback.

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

# A    TASKS

## A.1    MEANING SPACE

We assume that language is evolved and acquired only after other skills are mastered. In particular, we assume that agents already conceptualize the world in terms of objects with properties. This seem like a reasonable assumption, given what we know about individuation in infants (Xu & Carey, 1996) and apes (Mendes et al., 2008).

Each meaning contains one or several objects with several properties, which we represent hierarchically using filler-role pairs. In turn, these discrete representations can be embedded and represented as sets of vectors used by neural architectures. In the future, such representations could be obtained directly using perceptual neural architectures which segment the inputs into objects and categorize each object's representation along several axes, yielding properties.

We will abbreviate the reconstruction task by REC and the discriminative task by DIS. The REC task will be described in depth in another publication, so we omit the details here. Instead, we focus on their similarities.

We first define a set of roles $R_0$ for the properties of objects. For example, $R_0$ can be $\{\text{COLOR}, \text{SHAPE}, \ldots\}$. For each role, there is a corresponding set of fillers, for example, the fillers corresponding to COLOR are $\{\text{BLUE}, \text{RED}, \ldots\}$. In DIS, these roles are arbitrary and meaningless but in REC, they have a clear interpretation. An object $o \in \mathcal{O}$ is a set of filler-role pairs such as a blue circle, represented as $o_1 = \{(\text{COLOR}, \text{BLUE}), (\text{SHAPE}, \text{CIRCULAR})\}$. In Figure 1, each object is represented by a row of 3 integers, where each column correspond to a role and each integer to a filler for that role.

In both tasks, the inputs contain several objects, which will be indexed by different sets of roles. In DIS, we define $R_1 = \{1, 2, 3, \ldots\}$, which could be interpreted as a position in space and embedded using Cartesian coordinates. In REC, the roles $R_1 = \{\text{AGENT}, \text{PATIENT}, \text{MISC}\}$ correspond to $\theta$-roles. An object of the meaning space $\mathcal{X}$ is a set of filler-role pairs, where the filler space is $\mathcal{O}$ and the role space is $R_1$.

In DIS, the meaning $x \in \mathcal{X}$ is observed by both speaker and listener. One of the pair $(r, o) \in x$ plays the special role of *target* whil eother pairs are *distractors*. The target should be communicated to the listener to succeed the task. We call the listener's observations the *context*. In DIS, the context is simply the meaning $x$.

In REC, the meaning is fully observed by the speaker, but only partially observed by the listener. In other words, the context is $c \subsetneq x$. The information to convey by the speaker to the listener is every part of the meaning that is not part of the context, i.e. the complementary of $c$ in $x$.

## A.2    SIMULATING PRAGMATIC INFERENCES

We further assume that agents follow the Gricean maxim of quantity (Grice, 1975): agents communicate exactly the information that is needed for solving the task, not more, not less. In REC, there is no need to send the entire meaning to the listener, but the complementary of the context. In the DIS task, there is a minimal set of pairs that uniquely distinguish the target from the other objects.

Ideally, the agents should be able to perform pragmatic inferences themselves. Indeed, the maxim of quantity probably stems from a simpler objective trading off communication efficiency and the efforts of the agents, as formulated by Zipf (1949) and similar to the objective that we use (presented below). However, as a first step towards this more realistic setup, the pragmatic inference that determine what to send is given to the senders. In REC, the objects that are not observed by the listener are indicated to the sender by a specific flag, which is embedded and added to the object representations. In DIS, the properties of the target are embedded and given to the sender, ignoring the distractors. The minimal set of properties to identify the target are identified with a specific flag as well. We are currently working on architectures which would compute this minimal set of properties by observing all the objects. In Figure 1, we represent these pragmatic inferences as boolean indicators that are observed by the speaker.

Importantly, some least-effort pressure is necessary for the speaker to only send information that is marked as pragmatically relevant. Without this pressure, if the message space is large enough (if the vocabulary size and the maximum message length are large enough), the sender could simply represent its entire input (meaning and flags). The least-effort pressure should encourage the sender to leverage these pragmatic markers.

## A.3 LOSS

Let $(x, t)$ be a meaning along with a target. The context $c$ is uniquely determined by $(x, t)$. The speaker observes $(x, t)$ and produces a message $e \sim q(\cdot|x, t)$ which is then transmitted to the listener. The loss function is composed of an information transmission term and a least-effort penalty,

$$\mathcal{L}(x, t, e) = l(t, e) + r(e, x, t).$$

**Information transmission term:** In the discriminative case, the listener produces a conditional probability $p$ over the possible targets $\mathcal{T}$, and the information transmission term to minimize is

$$l(t, e) = -\log p(t|e, c).$$

In the reconstruction case, the listener produces a conditional probability over the meaning space $\mathcal{X}$, and the information transmission term is

$$l(t, e) = -\log p(x|e, c).$$

Recall that in REC, $x \in \mathcal{X}$ is a set. The distribution over this set is obtained by predicting for each role $r \in R_1$ all the features of all the properties $r' \in R_0$, as well as predicting whether $(o, r) \in x$. That is, $p(x|e, c) = \prod_{r \in R_1} p(o_r|r, e, c)p(\gamma_r|r, e, c)$ where $\gamma_r$ is a binary random variable indicating whether there is a pair in $x$ containing the role $r$.

**Least-effort penalty:** The second term, $r(e, e, t)$, penalizes the speaker for using long messages. As a first approximation, the length of the message is proportional to the energy spent by the speaker to produce the message. In Chaabouni et al. (2019), the function $r$ was defined as

$$r(e) = \lambda|e|,$$

where $\lambda$ is a hyperparameter, and $|e|$ is the number of symbols (except the end-of-sentence symbol) in the message.

However, we noticed that messages collapse to empty messages early on during training. This is similar to the well-known *posterior collapse*, where the approximate posteriors of latents of sequence-to-sequence VAEs collapse to their priors (Bowman et al., 2016). We fix the issue by adapting two well-known tricks: Pelsmaeker & Aziz (2020)'s minimum desired rate and Kingma et al. (2016)'s free bits. The penalty term becomes

$$r(e, x, t) = \mathbf{1}_{l(t,e)<\tau}\mathbf{1}_{|e|>n_{min}}(\lambda|e|),$$

where $\mathbf{1}$ is the indicator function.

For this term to be non-zero, two conditions need to be fulfilled. Firstly, the reconstruction error must be below $\tau$, which is analogous to a minimum desired rate. The interpretation is that the speakers only start minimizing their efforts once they have managed to get their point across. Secondly, the penalty is above 0 only if the message contains more than $n_{min}$ symbols. This gives models $n_{min}$ "free" symbols for each datapoint. Without this factor, we found that speakers often utter empty messages.

# B  DISCRIMINATIVE TASK

In this section, we describe the data used in the DIS task and the architecture of the agents.

## B.1  DATA

The data is synthetic. Each role $R_0$ is also called a property or feature since it characterizes an object and is numbered from 1 to $n_f$. Each object is composed of all roles from $R_0$ and a filler for that role which is simply number from $\{1, \ldots, n_v\}$. Thus each object can be represented easily as a dense vector. In our experiments, we use $n_v = 8$ and $n_f = 5$.

The data is generated as follow. First, we draw the number of objects $N$ from a categorical distribution $p_k = P(N = k)$ defined by $p_2 = p_3 = \frac{1}{8}$, $p_4 = \frac{1}{8}$, $p_5 = \frac{1}{4}$. Then, the first object object $u_1$ is obtained by sampling a value uniformly from $\{1, \ldots, n_v\}$ for each feature. The next object $u_2$ is obtained by sampling a feature $f_1 \in R_0$ and resampling a different value. $u_3$ is obtained by sampling a different feature $f_2 \in R_0 \{f_1\}$ and by resampling another value, etc.

This process has an interesting property: the first and the last objects are distinguishable from all the other objects by a single feature while the other objects are distinguishable from all the other objects by two features. We call these features the *necessary* features, denoted by the random variable $K$. This is illustrated in Table 2.

Furthermore, in order to create an out-of-distribution (*OoD*) test set, we reject some datapoints. When $K = 2$, we reject datapoints where feature 1 is necessary, and we also reject datapoints where features 2 and 3 are necessary (but feature 2 and 3 can be necessary features along with another feature). As a result of the distribution of $N$ and the OoD rejection procedure, we obtain $P(K = 1) \approx 0.69$. We have not yet tested our models on these test sets but intend to do so.

$$
\begin{aligned}
u_1 &= [\quad 4\,, 2\,, 1\,, 5\,, 3\quad] \\
u_2 &= [\quad 4\,, 5\,, 1\,, 5\,, 3\quad] \\
u_3 &= [\quad 4\,, 5\,, 1\,, 3\,, 3\quad] \\
u_4 &= [\quad 4\,, 5\,, 1\,, 3\,, 1\quad] \\
u_5 &= [\quad 4\,, 5\,, 6\,, 3\,, 1\quad]
\end{aligned}
$$

Table 2: Illustration of the data generation process. Objects are generated sequentially, starting from a uniformly random vector $u_1$, where each position in the vector corresponds to a role from $R_0$. $u_{i+1}$ is obtained by modifying a single feature (underlined) from $u_i$, a feature that was never modified previously. Thus each object is uniquely identifiable from the set of objects, using either 1 necessary feature ($u_1$ and $u_5$) or 2 necessary features ($u_2, u_3, u_4$) (necessary features to identify $u_i$ are colored on the corresponding line).

## B.2  MODEL

In order to use neural models, we represent the inputs $x$ and contexts $c$ as tensors. In all our architectures, attention mechanisms (Bahdanau et al., 2014) plays the crucial role of selecting what information to send next. In particular, the models used are based on Transformers (Vaswani et al., 2017). The main difference between the architectures in the two tasks is that the sender attends over objects in the argument structure setup, while it attends over properties of the target object in the intersective adjective setup.

In DIS, the sender observes a representation of the target object as a set of properties over which it can attend. The necessary properties (that are pragmatically relevant) are marked. Formally, for each pair of the target object $(r, f)$ corresponds a row $V_i$ in the matrix $V$, defined as

$$
V_i = \text{Val}_r(f) + \text{Role}(r),
$$

where $\text{Role}$ and $\text{Val}_r$ are learned embedding matrices. This matrix $V$ is attended over by a Transformer decoder, which predicts log probabilities over the vocabulary auto-regressively, via causal masking. We use the Gumbel-Softmax straight-through (Jang et al., 2017; Maddison et al., 2016) to backpropagate through the discrete decisions.

The listener is made of two parts. Firstly, a Transformer encoder processes the message and produces a sentence representation $z$ by linearly transforming the contextualized end-of-sentence embedding. Secondly, each object $o_i = \{(r_1, f_1), \ldots, (r_n, f_n)\}$ is represented vectorially as

$$u_i = W[\text{Val}'_{r_1}(f_1); \ldots; \text{Val}'_{r_n}(f_n)] + b,$$

where $[\cdot; \cdot]$ denotes the concatenation of vectors, $\text{Val}'_r$ is an embedding matrix specific to the role $r \in R_0$ and $W$ is a matrix that lowers the dimensionality of the vector from $d \times n$ to $d$. Finally, the probability of the object $i$ being the target is simply $p(t = i|e, c) \propto z^T u_i$.

## C    EXPERIMENTAL SETUP

We perform a random search (Bergstra & Bengio, 2012) over a variety of hyperparameter values. The agents of interest use non-empty messages, but also messages which length vary as is the case in natural language. Thus, we tune the hyperparameters to achieve this goal. We set the maximum length to 8 symbols (excluding the end-of-sentence token) and filter out runs which on average use less than 1 symbol or more than 6 symbols.

The following values are specific to the DIS task. We use the Adam optimizer (Kingma & Ba, 2014) with a learning rate of 0.0003, with $\beta_1 \sim U(\{0.3, 0.9\})$, $\beta_2 \sim U(\{0.9, 0.99, 0.999\})$ and the batch size follows $U(\{32, 128\})$. The training, validation and test set are all drawn from the same distribution and contain 3000, 1000 and 1000 examples each. We validate every 5 epochs and compute the loss. If the loss hasn't improved for 15 validations in a row, we stop the optimisation. The temperature of the Gumbel-Softmax estimator lies in $\{0.9, 1.0, 1.2\}$. As for the loss, the number of free symbols is either 0 or 1, $\lambda \sim U\{0.1, 0.3, 1.0\}$, and $\tau \sim U\{0.1, 0.3\}$. Finally, the vocabulary size is drawn from $U(\{32, 128\})$.

We use the EGG framework (Kharitonov et al., 2021) to run our experiments and in particular, PyTorch (Paszke et al., 2019) for implementing our models. The dropout parameter is drawn from $\{0.2, 0.3\}$. The sender's Transformer can have 1, 2 or 3 layers and similarly for the message encoder of the Listener.

## D    RESULTS

### D.1    CONCATENABILITY AND QUALITATIVE ANALYSIS

Concatenability metrics are information-theoretic quantities measured in nats. When $C^L = -0.05$, on average, the cross-entropy loss incurred by the listener is 0.05 higher when it receives the best concatenated messages ($e_1 e_2$ or $e_2 e_1$) compared to when the message produced by the speaker using greedy decoding $e_{1,2}$ is received. The loss of information of 0.05 is very small: in comparison, the loss incurred by a random baseline in a discriminative task with uniform distribution over two objects that are the targets with equal probability is $\log 2 \approx 0.693$. On the other hand, 1.19 is rather large for a task with at most 4 distractors, since it is above $\log 3 = 1.09$. Thus, the two models we have selected have vastly different concatenability metrics $C^L$.

$C^S$ is in general a much bigger quantity, since it is defined relatively to the log probabilities of messages, which live in a much bigger space. $C^S$ and $C^L$ are very correlated (Spearman coefficient of 0.63). This is not very surprising since the sender and the listener are trained jointly. In particular, **B** also has the highest $C^S = -5.66$ among the runs, meaning that the concatenated messages (obtained systematically) have a probability relatively close to the actually sent messages.

| Meaning | A | B |
|---|---|---|
| $-,1,-,-,-$ | 3 | 13 |
| $-,-,-,1,-$ | 24, 24, 24 | 12 |
| $-,1,-,1,-$ | 24 , 3 , 24 | 13 , 12 |
| $-,1,-,-,-$ | 3 | 13 |
| $-,-,-,2,-$ | 11 | 22 |
| $-,1,-,2,-$ | 11 | 6, 29 |
| $-,1,-,-,-$ | 3 | 13 |
| $-,-,-,-,1$ | 27 | 13 |
| $-,1,-,-,1$ | 3 | 13 |
| $-,-,-,1,-$ | 24, 24, 24 | 12 |
| $-,-,-,-,1$ | 27 | 13 |
| $-,-,-,1,1$ | 24 , 27 , 24 | 13 , 12 , 13 |
| $-,-,-,1,-$ | 24, 24, 24 | 12 |
| $-,-,-,-,2$ | 31, 31 | 6 |
| $-,-,-,1,2$ | 11, 24 , 30, 11 | 6 , 12 |
| $-,-,-,2,-$ | 11 | 22 |
| $-,-,-,-,1$ | 27 | 13 |
| $-,-,-,2,1$ | 24 | 22 , 13 |
| $-,-,-,2,-$ | 11 | 22 |
| $-,-,-,-,2$ | 31, 31 | 6 |
| $-,-,-,2,2$ | 11 , 9 | 6 |
| $-,1,-,-,-$ | 3 | 13 |
| $-,-,-,-,2$ | 31, 31 | 6 |
| $-,1,-,-,2$ | 31 , 3 , 31 | 6 , 13 |

Table 3: Most frequent messages from the models with the lowest $C^L$ (**A**: $C^L = -1.19$ on train set) and highest (**B**: $C^L = -0.05$) in the DIS task, for some random selection of meanings. Meaning: sequence, where position denotes the property $R_0$ and number the corresponding filler. Symbols are colored to identify whether messages for complex meanings include symbols used to convey simple meanings. Model **B** has high concatenability and 4/8 complex messages are obtained by concatenating simple messages, compared to 0/8 for the **A** model.

