# OpenReview forum: "Varying meaning complexity to explain and measure compositionality"
_ICLR.cc/2022/Workshop/EmeCom — EmeCom Workshop at ICLR 2022_

### Official Review · Reviewer_7wSL · 2022-03-21
**An experimental paper that lacks essential details about the setting but that introduces an interesting measure of compositionality that should be explored further**

**Rating:** Weak rejection
**Confidence:** 4

**Review:**

This paper proposes two different setups to measure compositionality by introducing "the meaning complexity variation" idea. That is, unlike prior works that train emergent languages relying on one fixed meaning complexity, this paper trains agent to communicate about meaning with different complexities (simple and complex). This idea allows the introduction of two measures of compositionality (listener's and speaker's concatenability). The main contribution of this paper is the concatenability measure. However, the latter has been only introduced and not analyzed. In the main paper, only the definition of the measure is given and only in the appendix, some intriguing results were given where both measures C^L and C^S give a non-coherent picture.
Furthermore, the authors motivate the 2 introduced setups by their realism compared to existing works, which does not seem the case. For example, in the first introduced setup, the attributes that the speaker should describe are given. Authors argue that this is what happens in real life as the speaker can communicate only about a subset of the attributes. If the latter is true, it is for the speaker to decide if it wishes to communicate about only a part of the input and is not given directly. Hence, I think such claims about realistic setup should be revisited.
Finally, the paper lacks crucial details to interpret and reproduce the paper, such as more details about the input space, the architecture, the number of distractors, the channel capacity ...
In sum, I think the concatenability measures are a very strong addition to the community. However, these measures give contradicting results that should be explained further. Moreover, the paper lacks many details to be able to interpret the results.

Some specific questions to authors:
1. Does the speaker have access to the listener's inputs in the second introduced setup? If not, how does the game work?
2. In appendix: what do you mean by "with average reconstruction loss is 0.59". Is the training loss is a discriminative loss? If so how many distractors are you using? Is such a low loss can explain the difference between C^{L} and C^{S}? Adding a random baseline to study Listener's concatenability would make these results a bit more interpretable.

---

### Official Review · Reviewer_pJS1 · 2022-03-23

**Rating:** Weak accept
**Confidence:** 4

**Review:**

Summary of the contributions:

This paper explores the concept of compositionality in (emerging) languages. Firstly, it proposes two metrics to measure it (when defined with a reliance on homomorphism), speaker concatenability (CˆS) and listener concatenability (CˆL), which accounts, respectively, for the tendency of the speaker to speak ‘productively’ (in a way that shows systematicity - as ‘productivity’ and ‘systematicity’ are discussed in Hupkes et al. [2019], as far as I understand), and for the tendency of the listener to generalize compositional. Secondly, using Transformer-based architectures [Vaswani et al., 2017] for some speaker and listener agents, the paper reports on some measure of compositionality against those novel metrics in the context of novel referential-game-like tasks.

It takes inspiration from Zipf [1949]’s least-effort principle and advocates that there is a lack of realism in most settings/tasks where language emergence is studied, especially in the modern-era of deep learning. To wit, the paper makes the assumption that, during language emergence, grammatical structures grow in complexity to accommodate for the variation in complexity in the meanings that are attempted to be conveyed by the population of agent wielding said language. Therefore, it is important that the settings/tasks in which
language emergence is studied account for this “meaning complexity variation”.

Thus, the paper develops two tasks that attempt to fix that gap. One seek to promote emergence of intersective modifiers/adjectives, and the other is interested in a part of speech structure: the argument structure.

In appendix, the paper provides preliminary results.


Novelty, relevance, significance:

1) The ideas developed in the paper are original, as far as I can say.

2) The paper grapples with the concept of compositionality in emergent languages and provides a novel viewpoint, it is therefore very relevant to the community of the workshop, but it is maybe not so topical with this year’s push towards interdisciplinary collaborations.

3) This paper is providing a very significant viewpoint to the community of the workshop but the paper is not complete, as it lacks some results and analysis. Moreover, given the context of this year’s workshop, i.e. aiming to foster inter-disciplinary discussions, I am afraid that the paper, as it stands, falls short of that goal, as it is a very dense and rather difficult read (it might be making a lot of assumption about the reader’s background or may go too much in depth with regards to some points, e.g. when discussing the lack of realism in language emergence settings ).

I would advise the authors to make place for some results and their analysis, thus enhancing the completeness of this workshop-purposed paper, by maybe retrograding to the appendices the whole discussion around the state of the most language emergence tasks and how the proposed tasks fix any gap, and steer the focus of the paper on the compositionality metrics contribution, maybe?


Soundness:

I can recognise two claims in the paper. To wit, that (i) the proposed tasks provide more realism to the settings in which language emergence is studied, and that (ii) meaning complexity variation is a promising functional explanation for the emergence of compositionally.

Claim (i) is certainly difficult to evaluate and substantiate. I think the paper would benefit from reframing it as an assumption for the work presented.

Claim (ii) is mainly substantiated via Table 1 in the main text, and via Table 2 in the appendix. These are mainly qualitative evaluations. Measures of speaker and listener concatenability are also reported, as quantitative evaluations, but, as it stands, they do not show a clear picture of what is happening, as the claim nuanced (’promising’). I would advise the authors to attempt to evaluate the compositionality of the emerging languages using topographic similarity [Brighton and Kirby, 2006], , and posdis and bosdis [Chaabouni et al.,2020], if they can be applied (?).


Quality of writing/presentation & Literature:

(1) The paper is rather well-organised but there is not enough details in explaining training/implementation setup, making it difficult to reproduce and build upon.

(2) In terms of the literature, I think it would be valuable to position the paper with respect to the decomposition of the concept of compositionality as done in Hupkes et al. [2019] (as I have tried to highlight it in my summary above).
Also, the relationship with the work on positive signaling and positive listening in Lowe et al. [2019] (and Eccles et al. [2019], Lin et al. [2021]) might deserve to be fleshed out further. Indeed, as the concatenability metrics encompasse both linguistic (speaker one) and behavioural (listener one) aspects, I think that this work as the potential to make a bridge between the previously mentioned pieces of literature (and maybe also with the main result of the work of Chaabouni et al. [2020]). Such a bridge could be a very valuable synthesis effort for the community.


Decision:

Given the potential of this paper, I am evaluating it as a “weak accept” and would hope to raise it to an “accept” if the paper can be made more complete, by (i) by providing sufficient implementation details for anybody to build upon the presented work, and (ii) integrating into the main text more quantitative results and a proper discussion of their impact.


Some ideas that I was considering during my read, but I am unsure where to make them fit in this review (sharing them with hope that it can be helpful) :

•Generalisation tests are not considered, as results are shown on a training set and there is no mention of a splitting strategy. Yet, following the results of Chaabouni et al. [2020], for this paper to be complete and significant to the community, it seems important to provide a measure of the generalisation abilities of the agents trained in this new setting that is separate from the measure of compositionality, as it is unclear whether the compositional behaviours (as measured in the sense of CˆL) that the agents exhibit at training time, in this new referential-game-like setting, do indeed translate into similar compositional behaviours at testing time, when prompted with zero-shot compositions of attribute-value pairs.


•Following the last sentence in the appendix: The first part of this statement (”concatenations are on average rather well interpreted by the listtener”) seems to echo a result in Chaabouni et al. [2020]: in a different settings (reconstruction game), it was found that languages that were compositional in the sense of the posdis metric were more likely to lead to high generalization capabilities from the agents (as a behaviour, mainly exhibited by the listener agent). Thus, the results found here seems to provide a reason why compositionality in the sense of posdis in emerging languages favours generalization capabilities, i.e. that the listener agents with such architectures may have some kind of natural propensity towards posdis-compositional languages.

The second part of the statement (”[concatenations] are not likely according to the speaker”) seems to highlight the extent with which modern-day neural network architectures that we use to model speaker agents do not have induction biases that favours the emergence of compositionality (as we already had some clue from Kottur et al. [2017]). I think those results would call for subsequent experiments involving an agent following the Obverter approach [Choi et al., 2018, Batali, 1998], for it sees the speaker agent outputing messages via a mechanism tied to its possible role as a listener agent, thus I would expect the measure of CˆS on such agent to be high enough to show that concatenations are likely according to obverter-like speakers.


References:

J. Batali. Computational simulations of the emergence of grammar. Approach to the Evolution of Language, pages 405–426, 1998.

H. Brighton and S. Kirby. Understanding Linguistic Evolution by Visualizing the Emergence of Topographic Mappings. Artificial Life, 12(2):229–242, jan 2006. ISSN 1064-5462. doi: 10.1162/artl.2006.12.2.229. URL http://www.mitpressjournals.org/doi/10.1162/artl.2006.12.2.229.

R. Chaabouni, E. Kharitonov, D. Bouchacourt, E. Dupoux, and M. Baroni. Compositionality and Generalization in Emergent Languages. apr 2020. URL http://arxiv.org/abs/2004.09124.

E. Choi, A. Lazaridou, and N. de Freitas. Compositional Obverter Communication Learning From Raw Visual Input. apr 2018. URL
http://arxiv.org/abs/1804.02341.

T. Eccles, Y. Bachrach, G. Lever, A. Lazaridou, and T. Graepel. Biases for emergent communication in multi-agent reinforcement learning. Dec. 2019.

D. Hupkes, V. Dankers, M. Mul, and E. Bruni. Compositionality decomposed: how do neural networks generalise? aug 2019. URL
http://arxiv.org/abs/1908.08351.

S. Kottur, J. M. F. Moura, S. Lee, and D. Batra. Natural Language Does Not Emerge ’Naturally’ in Multi-Agent Dialog. jun 2017. URL
http://arxiv.org/abs/1706.08502.

T. Lin, M. Huh, C. Stauffer, S.-N. Lim, and P. Isola. Learning to ground Multi-Agent communication with autoencoders. Oct. 2021.

R. Lowe, J. Foerster, Y.-L. Boureau, J. Pineau, and Y. Dauphin. On the Pitfalls of Measuring Emergent Communication. mar 2019. URL
http://arxiv.org/abs/1903.05168.

A. Vaswani, N. Shazeer, N. Parmar, J. Uszkoreit, L. Jones, A. N. Gomez, L. Kaiser, and I. Polosukhin. Attention is all you need. Advances in neural information processing systems, 30, 2017.

G. K. Zipf. Human behavior and the principle of least effort. 1949.

---

### Decision · Program_Chairs · 2022-03-25

**Decision:**

Accept

**Comment:**

This paper proposes an interesting scenario of different levels of complexity and a new measure of compositionality. Reviewers have strong concerns about experimental details as well as depth of analysis and experiments. Given the potential of the author to add these details (maybe release code as well), add new experiments, and update their paper, we are accepting this work on the merit of the idea and initial experiments. We highly encourage the author to make the suggested changes before the workshop, and hopefully there will be new results that may be even more interesting to discuss.